# Prognostic Abilities of Pre- and Post-Treatment Inflammatory Markers in Oral Squamous Cell Carcinoma: Stepwise Modelling

**DOI:** 10.3390/medicina58101426

**Published:** 2022-10-10

**Authors:** Sarah Sabrina Zakaria, Anand Ramanathan, Zakiah Mat Ripen, Wan Maria Nabillah Ghani, Yi-Hsin Yang, Vui King Vincent-Chong, Siti Mazlipah Ismail

**Affiliations:** 1Department of Oral & Maxillofacial Clinical Sciences, Faculty of Dentistry, Universiti Malaya, Kuala Lumpur 50603, Malaysia; 2Oral Cancer Research & Coordinating Centre, Faculty of Dentistry, Universiti Malaya, Kuala Lumpur 50603, Malaysia; 3National Institute of Cancer Research, National Health Research Institutes, Tainan 70456, Taiwan; 4Department of Oral Oncology, Roswell Park Comprehensive Cancer Center, Buffalo, NY 14263, USA

**Keywords:** inflammatory markers, oral squamous cell carcinoma, prognosis, overall survival, disease-free survival

## Abstract

*Background and Objectives:* Studies examining the importance of inflammatory markers before treatment as prognosticators of OSCC are available, but information on post-therapy inflammatory markers and their prognostic significance is limited. This study aimed to evaluate the prognostic abilities of pre- and post-treatment inflammatory markers in patients with OSCC. *Materials and Methods:* In this retrospective analysis, information on 151 OSCC patients’ socio-demographic, clinico-pathological, recurrence, metastasis, and survival data were gathered from clinical records. A multivariable Cox proportional hazards regression (stepwise model) was conducted to identify the prognostic predictors of OS and DFS. The multivariable models’ performances were evaluated using Harrell’s concordance statistics. *Results:* For OS, high pre-treatment LMR (HR 3.06, 95%CI 1.56, 5.99), and high post-treatment PLC (HR 3.35, 95%CI 1.71, 6.54) and PLR (HR 5.26, 95%CI 2.62, 10.58) were indicative of a poor prognosis. For DFS, high pre-treatment SII (HR 2.59, 95%CI 1.50, 4.48) and high post-treatment PLC (HR 1.92, 95%CI 1.11, 3.32) and PLR (HR 3.44, 95%CI 1.98, 5.07) were associated with increased mortality. The fitness of the OS and DFS stepwise Cox regression models were proven with a time-dependent AUC of 0.8787 and 0.8502, respectively. *Conclusions:* High pre-treatment levels of LMR and SII and high post-treatment levels of PLC and PLR are independent predictors of a poor prognosis for patients with OSCC.

## 1. Introduction

It is estimated that there were more than 370,000 new cases and more than 170,000 deaths reported worldwide in 2020 for oral cavity and lip cancer [1]. In Malaysia, oral cancer is one of the ten most commonly occurring cancers among the Indian ethnic group [2]. The most common treatment modalities for head and neck cancers are surgery, radiotherapy and chemotherapy. However, despite effective treatment efforts, the prognosis for oral cancer patients is poor, with a 5-year survival rate of approximately 50% [3,4]. The pathological tumour-node-metastasis (pTNM) stage is currently considered as the best predictor of disease progression and long-term survival; however, pTNM staging is not available prior to surgery [5].

Inflammation has been accepted as one of the hallmarks of cancer. Over the years, there has been growing interest regarding the relationship between inflammation and tumour microenvironment. Studies have been conducted to assess the roles of inflammatory markers as prognostic indicators for cancer progression. It has been documented that the systemic inflammatory response promotes tumour microvascular regeneration, tumour metastases and tumour cell proliferation [6], and it is hypothesised to be represented by inflammatory markers such as neutrophils, lymphocytes, monocytes and platelets.

More recently, several studies have proposed various scoring systems using these inflammatory markers, with the most commonly proposed and studied being the neutrophil–to–lymphocyte ratio (NLR), the platelet–to–lymphocyte ratio (PLR) and the lymphocyte–to–monocyte ratio (LMR). These scoring system markers have been shown to be good prognostic indicators for various types of malignancies. A meta-analysis has concluded that NLR and PLR are useful prognostic indicators for ovarian cancer [7]. Similarly, meta-analyses conducted among patients with non-small cell lung cancers [8] and oesophageal cancers [9] have found that high pre-operative levels of NLR and PLR are associated with a poor prognosis.

Several studies on the efficacy of these systemic inflammatory markers as prognostic indicators have also been conducted among patients with oral squamous cell carcinoma (OSCC). A recent study, conducted among patients who had undergone surgical resection, found that pre-operative NLR is an independent predictor of prognosis and could be used as an auxiliary parameter for overall survival (OS) and disease-free survival (DFS) before curative surgery is conducted [10]. Other studies conducted among patients with OSCC, further supported the role of pre-operative NLR, PLR and LMR as robust predictors for OS and DFS [11,12].

Although there are studies evaluating the prognostic abilities of these systemic inflammatory markers among various malignancies, predominantly, these studies have looked into the role of these markers at the pre-treatment level. In addition, there is a lack of information with regard to post-treatment inflammatory markers and their role in predicting the prognosis of cancer patients. Furthermore, most of these studies have reported on the more well-known markers, such as PLR and NLR [8,9,10,11]. There are scarce data on the prognostic abilities of other systemic inflammatory markers scoring systems, such as the lymphocyte–to–white blood cell ratio (LWR), the white blood cell–to–haemoglobin ratio (WHR), the derived neutrophil–to–lymphocyte ratio (dNLR) and the systemic immune-inflammation index (SII). As such, the prognostic abilities of these potentially important markers cannot be determined, thus, warranting further studies aimed at evaluating the prognostic abilities of these various inflammatory markers and their scoring system.

Therefore, the objective of this retrospective study was to evaluate the prognostic abilities (in terms of OS and DFS) of various systemic inflammatory markers at both the pre- and post-treatment level, among patients diagnosed with cancers of the oral cavity. It is hypothesized that systemic inflammatory markers at both the pre- and post-treatment level have the potential to be used as prognostic indicators for patients diagnosed with OSCC.

## 2. Materials and Methods

### 2.1. Study Design and Population

This retrospective study was approved by the Medical Ethics Committee, Faculty of Dentistry (FOD), Universiti Malaya (UM) [MEC:OM1902/0026]. A total of 316 records of patients who were diagnosed with OSCC at the Department of Oral and Maxillofacial Clinical Sciences (OMFCS), FOD, UM, from 1 June 2000 to 31 December 2020, were evaluated for this study. Only patients with confirmed histopathological examination of OSCC and with complete data, including clinical information and blood analysis, were included (*n* = 153) (Figure 1). Patients with autoimmune diseases such as systemic lupus erythematosus, those with haematological disorders such as idiopathic thrombocytopenia purpura, and those with prolonged usage of corticosteroid therapy (*n* = 2) were also excluded. After excluding these patients, only 151 patients were eligible for inclusion in this study.

### 2.2. Data Collection

Socio-demographic data, such as age, gender, race and presence or absence of risk habits (smoking, alcohol consumption and betel quid chewing), as well as clinico-pathological data, such as pre- and post-treatment inflammatory markers in peripheral blood, site, clinical TNM staging, grading, treatment modalities, occurrence of recurrence/distant metastasis/second primary tumour, survival and follow-up status were collected from their clinical records. The time until events of recurrence, distant metastasis or a second primary tumour were also recorded. For patients who were lost to follow-up, a telephone call interview was made to determine the patient’s current survival status.

Inflammatory markers included in this study are shown in Table 1. Inflammatory marker measurements for both before and after treatment were extracted from patients’ clinical records. Pre-treatment haematological measurements were taken within 1 week of the start of treatment. Post-surgery haematological measurements that were taken at least 1 week after surgery were obtained to ensure that the wound healing process did not affect the results. Similarly, post-radiotherapy haematological measurements that were taken at least 1 week after radiotherapy, and post-chemotherapy measurements, which were taken at least 2 weeks after completion of chemotherapy, were obtained.

### 2.3. Statistical Analysis

Descriptive statistics were used to report the baseline characteristics of the study population. Categorical variables were recorded as frequencies and proportions (%). Numerical data were tested for normality of distribution using the Kolmogorov–Smirnov test and reported as mean and standard deviation. Changes between pre- and post-therapeutic inflammatory marker measurements were analysed using the Wilcoxon signed rank test, whereby effect size was calculated using Cohen’s *d*.

To identify significant factors for prognostic model, we first dichotomized all haematological inflammatory markers into two groups (high and low), based on the optimal cut-off values obtained from receiver operating characteristic (ROC) curves with Youden index. Only markers with area under the curve (AUC) value of >0.6 were selected for subsequent univariate and multivariable analyses.

OS was calculated from the date of diagnosis to the date of death or last follow-up, whereas DFS was calculated from the last date of primary treatment to the date of death, date of recurrence, distant metastasis, second primary tumour, or last follow-up. Survival curves for OS and DFS were generated using Kaplan–Meier method, whereby log-rank tests were used to compare differences between groups. Univariate and multivariate Cox proportional hazards regression analyses were then undertaken to identify prognostic predictors of OS and DFS, whereby the hazard ratio (HR) was calculated for each variable. Variables with *p* < 0.05 in the univariate analysis were entered into the multivariable Cox regression analysis (stepwise model) to assess its role as an independent prognostic marker. The Harrell’s concordance statistics were used to evaluate the performance of the multivariable models. Data analysis was conducted using SPSS for Windows (version 23.0) and SAS (version 9.4). Statistical significance was defined at a 2-tailed *p*-value of <0.05.

## 3. Results

A total of 151 patients, who had been histologically diagnosed with OSCC, were included in this study. Table 2 and Table 3 show the socio-demographic and clinico-pathologic profile of the patients. The mean age (±sd) for this study cohort was 59.70 (±13.87) years old, approximately two-thirds (62.9%) were females and some had co-morbidities (69.5%) such as hypertension and diabetes mellitus. Half of the cohort practiced oral cancer risk habits, with betel quid chewing being the most common habit (31.1%) seen in this cohort. Cancers of the tongue and the floor of the mouth accounted for the largest proportion (41.7%) of cases, with almost half of them being diagnosed at stage IV (48.3%). The majority of the patients underwent surgery (90.7%) as part of their treatment modality, half of them underwent radiotherapy (55.0%), and only a small proportion (15.2%) underwent chemotherapy. Recurrence was seen in 22.5% of the study population. At the point of the data analysis, one-third of the patients were deceased and 12.6% were lost to follow-up.

A comparison of level of inflammatory markers pre- and post-treatment is shown in Table 4. Except for AMC, WBC, dNLR, MWR and WHR, all the other inflammatory markers showed a significant difference between pre- and post-treatment. Most of the inflammatory markers showed a significant increase in their levels (ANC, PLC, NLR, PLR, MLR, NWR, PWR and SII) at post-treatment. The most significant increase was seen in PLR, with Cohen’s *d* value of 0.634. On the other hand, a significant decrease was seen in the levels of ALC, Hb, LMR and LWR at post-treatment.

Next, univariate and multivariable Cox regression analyses were conducted, using both the pre- and post-treatment inflammatory markers in the model to identify the markers that could be predictive of OS and DFS. Table 5 shows the variables that were associated with patients’ OS. An advanced disease stage was observed to be a significant predictor of poor OS (HR 2.42 95%CI 0.99, 5.87 *p* = 0.050), while those who underwent surgery were shown to have a lesser likelihood of mortality (HR 0.35 95%CI 0.14, 0.91 *p* = 0.031). High levels of pre-treatment LMR (HR 3.06 95%CI 1.56, 5.99 *p* = 0.001), and high levels of post-treatment PLC (HR 3.35 95%CI 1.71, 6.54 *p* = 0.000) and PLR (HR 5.26 95%CI 2.62, 10.58 *p* < 0.001) were indicative of a significantly poorer prognosis.

Analyses were also conducted to elucidate the markers that are predictive of DFS (Table 6). The role of PLC and PLR as prognostic indicators were again observed. High post-treatment PLC and PLR was shown to increase the likelihood of mortality by 2–3 folds (HR 1.92 95%CI 1.11, 3.32 and HR 3.44 95%CI 1.98, 5.07, respectively). In addition, high levels of pre-treatment SII were also found to be associated with a poor prognosis (HR 2.59 95%CI 1.50, 4.48).

Kaplan–Meier graphs for the OS and DFS of the markers that were identified as having significant prognostic abilities in the multivariable analyses are as attached in Appendix A.

The fitness of the models used in the analyses are illustrated in Figure 2 and Figure 3. Harrell’s concordance statistics showed that the model (which incorporates both pre- and post-treatment inflammatory markers) for OS was stable across time, with a time-dependent area under the curve (AUC) estimate of 0.8787 (Figure 2). Similar findings were observed for the DFS model, whereby the fitness of the model was also proven with a time-dependent AUC of 0.8502 (Figure 3).

## 4. Discussion

This study was conducted to evaluate the significance of the pre- and post-treatment inflammatory markers in the prognosis of OSCC patients, so as to assess their ability as prognostic markers. The post-treatment markers that were found to be significantly associated with prognosis were PLC and PLR, and LMR and SII were significant prognostic markers at pre-treatment. The stage of the tumour and surgery as a treatment modality were also found to be significant prognostic indicators.

In comparing the levels of these inflammatory markers, most of them were found to be increased after treatment; however, some markers demonstrated significantly lower levels post-treatment. The most significant increase in the levels between pre- and post-treatment were seen for PLR, whilst LMR showed the most significant reduction. Currently, there are not many studies that have looked into the dynamic changes in the pre- and post-treatment values of peripheral inflammatory markers in OSCC. A study in gastric cancer patients reported that changes in the values for LMR post-12-months is helpful in predicting long-term survival [13]. Apart from that, lymphocytes are one of the most radiosensitive cells and will be markedly decreased after radiotherapy [14].

In this study, we assessed the prognostic significance of systemic inflammatory markers, using models that include both pre- and post-treatment levels. High pre-treatment LMR and elevated post-treatment PLC and PLR were found to be independent indicators of poor OS, whereas for DFS, high post-treatment PLC and PLR, and high pre-treatment SII were found to increase the likelihood of a poor prognosis. Although the inflammatory response is generally considered to have recovered within the first week post-treatment, there may be some patients who have not fully recovered.

In the present study, PLC and PLR were the inflammatory markers that were found to be significantly associated with both OS and DFS. The significance of PLR as a prognostic indicator, as shown in the present study, is in concordance with the findings from previous studies. High PLR levels were documented as a significant predictor of mortality in a study conducted among patients with head and neck squamous cell carcinomas [15]. Tangthongkum et al. [16] compared the survival rates between OC patients with high and low PLR and observed that OS was significantly higher in the low PLR group. A recently conducted meta-analysis further illustrated the association between elevated PLR levels and poorer OS and DFS [17]. Apart from OC, elevated levels of PLR have also been shown to be associated with poorer OS and disease-/metastasis-free survival among other types of cancers, such as breast [18], gastric [19], ovarian [20] and oesophageal [21].

The association between PLR and prognosis can be explained by the fact that PLR is an index of increased platelets against decreased lymphocytes (high PLC). Platelets play a role in cancer progression as mediators in angiogenesis and immunomodulation, by secreting cytokines and growth factors that cause cell migration and proliferation. Platelet-derived TGF-β acts on cancer cells to activate the pathways promoting cancer metastasis. In addition, cancer cells produce mediators to further stimulate the production of platelets, thus producing a perpetual cycle that eventually leads to tumour progression [22]. Furthermore, it was thought that high PLC, increases the ability to create a thrombus involving the tumour cells, which migrates outside the blood vessels to create new tumoral beds at other sites, causing a relapse of the cancer [23].

Lymphocytes, on the other hand, play an important role in the host’s tumour defence. The adaptive immune system, in the presence of a tumour, responds by activating the lymphocytes to generate an effective anti-tumour cellular immune response, by activating cytotoxic T-lymphocytes, which are involved in killing cancer cells [24]. It has been suggested that a low lymphocyte count could be related to the dysfunction of immune surveillance against tumours, therefore, providing a convenient environment for metastasis and tumour progression [25]. Hence, having a high platelet count, against a low lymphocyte count (high PLR levels) explained its association with a poor prognosis for patients, as seen in the present study.

The present study also highlighted the significance of pre-treatment SII and LMR levels as a prognostic indicator for OS. The systemic immune-inflammation index (SII) is developed by combining peripheral neutrophils, lymphocytes and platelets to provide a better prognostic score. Neutrophilia and thrombocythemia may be associated with cancer progression and decreased survival [26]. On the other hand, lymphopenia indicates an immune deficit that may provide a favourable tumour environment [27]. A study by Diao et al. [28] identified SII as an independent prognostic factor for OSCC, as can also be seen in the present study, and found it to be superior to NLR and PLR in predicting survival rates. Elevated SII has also been indicated as a poor prognostic indicator for other types of cancer, such as breast [29] and colorectal [30].

With regard to LMR, Ong et al. [31], in a study conducted among early tongue cancer patients, concluded that low pre-treatment LMR is associated with poor survival, which concurs with the current study. Furthermore, low pre-treatment LMR was also seen to be associated with an increased tumour size, advanced nodal status and a higher pathologic state [32], which would lead to poor survival. LMR reflects the host’s immune status against the degree of tumour progression. Monocytes are able to migrate through the bloodstream into tissues and differentiate into macrophages that secrete various cytokines, which is involved in tumorigenesis, tumour progression and metastasis [32]. As such, a low lymphocyte and high monocyte count correlates to inadequate anti-tumour immunity and an increased tumour burden [33].

In assessing the prognostic ability of these inflammatory markers, we used the stepwise Cox regression modelling with Harrell’s concordance statistics to identify significant factors, while maintaining sufficient performance. The stepwise Cox regression OS and DFS model, incorporating both pre- and post-treatment markers, was found to be a fit and robust model for predicting the prognosis of OC patients, with an AUC of more than 0.8. This was a strength of this study. The limitations of this study were that it was a retrospective, single centre study. In addition, being retrospective in nature, there were many cases with incomplete inflammatory marker data, which had to be excluded. Hence, this could have introduced selection bias. Another limitation of this study was that the type and time duration of the surgeries were not considered for analysis. Surgeries that involve extensive reconstruction are likely to be lengthy and invasive. This would naturally lead to a longer period of higher or lower inflammatory marker values, which could have affected the results.

Therefore, it is recommended that a multi-centre prospective study, with a larger sample size be conducted. Such a study should take into account the type of surgical procedures and treatment being administered, the degree of invasiveness and the progression of the tumour, which would be reflected in the stage of the disease. This should increase the generalizability of the findings and eliminate any risk of bias. Findings from a large multi-centre study would enable a prognostic nomogram for OC patients to be produced, which could be applied universally in clinical settings to aid in the prognostication and better management of patients.

## 5. Conclusions

High pre-treatment levels of LMR and SII, and high post-treatment levels of PLC and PLR are independent predictors of a poor prognosis for patients with OSCC.These findings provide further evidence on the potential of inflammatory markers as prognostic indicators.This would help in risk stratification and an improved prognostication for patients with OSCC.

## Figures and Tables

**Figure 1 medicina-58-01426-f001:**
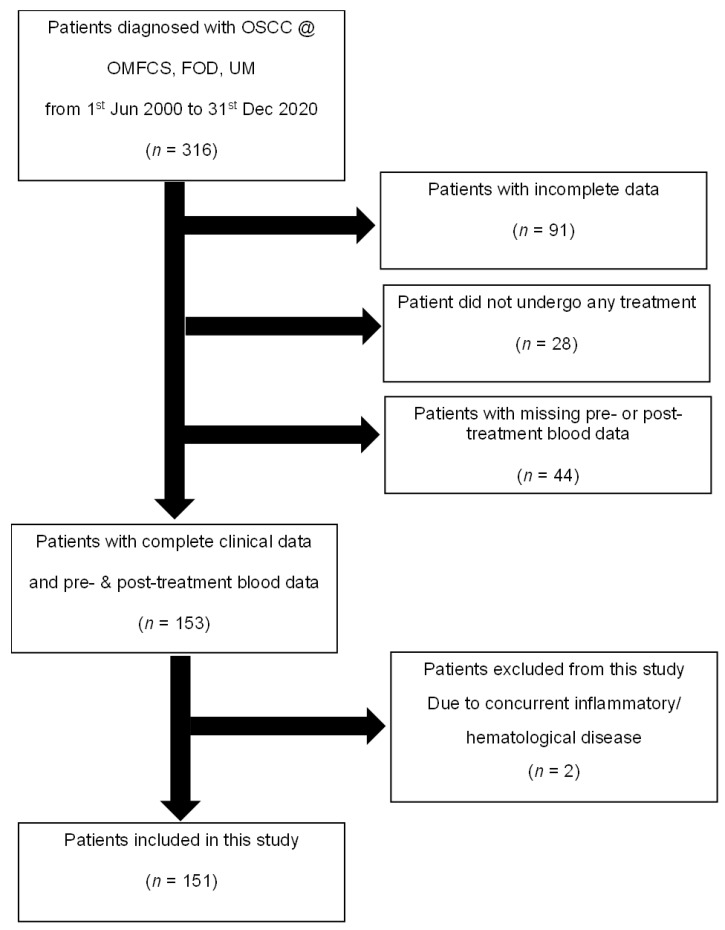
Flowchart explaining the patient selection process for this study.

**Figure 2 medicina-58-01426-f002:**
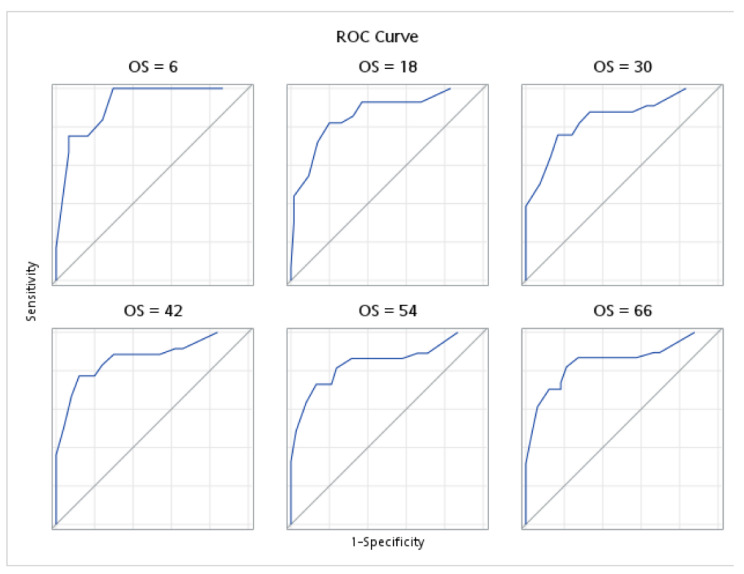
Harrell’s concordance statistics and time-dependent area under the curve for the overall survival (OS) in the pre- and post-treatment model.

**Figure 3 medicina-58-01426-f003:**
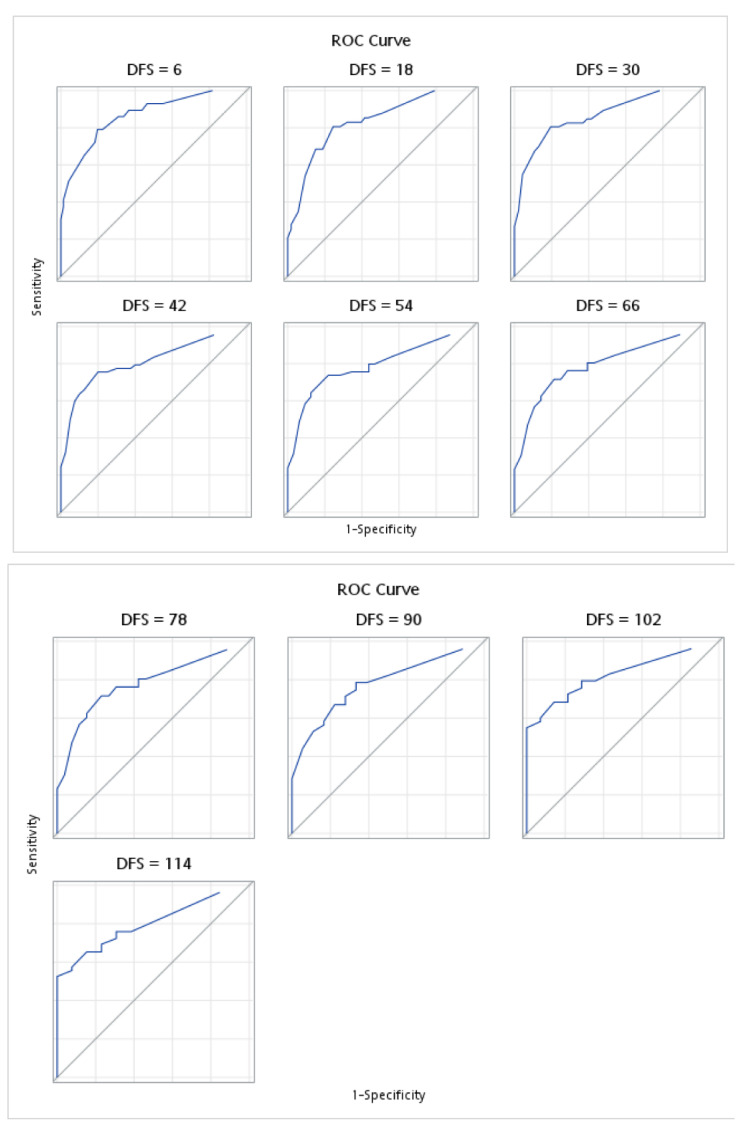
Harrell’s concordance statistics and time-dependent area under the curve for disease free survival (DFS) for pre- and post-treatment model.

**Table 1 medicina-58-01426-t001:** Inflammatory markers in peripheral blood included in this study.

Inflammatory Markers	Measurement
Absolute neutrophil count (ANC)	Number of neutrophils present in per microliter (×10^9^) of blood
Absolute lymphocyte count (ALC)	Number of lymphocytes present in per microliter (×10^9^) of blood
Absolute monocyte count (AMC)	Number of monocytes present in per microliter (×10^9^) of blood
Absolute platelet count (PLC)	Number of platelets present in per microliter (×10^9^) of blood
Haemoglobin (Hb)	Haemoglobin concentration in grams (g) per decilitre (dL) of blood
White blood cell (WBC) count	Number of white blood cells present in per microliter of blood
Neutrophil–to–lymphocyte ratio (NLR)	ANC/ALC
Platelet–to–lymphocyte ratio (PLR)	PLC/ALC
Lymphocyte–to–monocyte ratio (LMR)	ALC/AMC
Derived neutrophil–to–lymphocyte ratio (dNLR)	ANC/WBC–ANC
Monocyte–to–lymphocyte ratio (MLR)	AMC/ALC
Neutrophil–to–white blood cell ratio (NWR)	ANC/WBC
Lymphocyte–to–white blood cell ratio (LWR)	ALC/WBC
Platelet–to–white blood cell ratio (PWR)	PLC/WBC
Monocyte–to–white blood cell ratio (MWR)	AMC/WBC
White blood cell–to–haemoglobin ratio (WHR)	WBC/Hb
Systemic immune-inflammation index (SII)	PLC × ANC/ALC

**Table 2 medicina-58-01426-t002:** Socio-demographic characteristics of study population (*n* = 151).

Socio-Demographic Characteristics	n (%)
Age (years), mean ±SD	59.70 ± 13.87
<40 years old	17 (11.3)
≥40 years old	134 (88.7)
Gender	
Male	56 (37.1)
Female	95 (62.9)
Ethnicity	
Malay	21 (13.9)
Chinese	64 (42.4)
Indian	61 (40.4)
Others	5 (3.3)
Comorbidity	
Yes	105 (69.5)
No	46 (30.5)
Risk habit	
Yes	82 (55.0)
No	67 (45.0)
Smoking	
Yes	29 (19.2)
No	122 (80.8)
Alcohol	
Yes	26 (17.2)
No	125 (82.8)
Betel quid	
Yes	47 (31.1)
No	104 (68.9)

**Table 3 medicina-58-01426-t003:** Clinico-pathologic characteristics of study population.

Clinico-Pathologic Characteristics	n (%)
Site of tumour	
Tongue and floor of the mouth (ICD10 C01-2, 04)	63 (41.7)
Gingiva and palate (ICD10 C03, 05)	31 (20.5)
Buccal (ICD10 C06)	54 (35.8)
Lip (ICD10 C00)	3 (2.0)
Clinical TNM stage	
Stage I	26 (17.2)
Stage II	27 (17.9)
Stage III	25 (16.6)
Stage IV	73 (48.3)
Histologic differentiation	
Well-differentiated	55 (36.4)
Moderately-differentiated	92 (60.9)
Poorly-differentiated	4 (2.6)
Treatment modality	
Surgery	
Yes	137 (90.7)
No	14 (9.3)
Radiotherapy	
Yes	83 (55.0)
No	68 (45.0)
Chemotherapy	
Yes	23 (15.2)
No	128 (84.8)
Surgery only	68 (45.0)
Surgery, radiotherapy and chemotherapy	69 (45.7)
Radiotherapy and/or chemotherapy	14 (9.3)
Recurrence	
Yes	34 (22.5)
No	117 (77.5)
Survival status	
Alive	78 (51.7)
Deceased	54 (35.8)
Lost to follow-up	19 (12.6)
Follow-up range	1–217 months
	(median 30 months)

**Table 4 medicina-58-01426-t004:** Pre- and post-treatment comparison of inflammatory markers studied.

Parameter	Mean ± SD		Cohen’s *d*	*p*-Value
	Pre-Treatment	Post- Treatment		
ANC	5.94 ± 2.40	7.01 ± 4.02	0.270	0.005
AMC	0.72 ± 0.52	1.81 ± 8.46	0.128	0.179
PLC	292.04 ± 83.48	331.57 ± 122.71	0.356	<0.001
WBC	9.11 ± 2.90	9.87 ± 4.63	0.166	0.075
NLR	3.19 ± 2.02	7.02 ± 7.36	0.549	<0.001
PLR	153.79 ± 73.72	309.74 ± 255.03	0.634	<0.001
MLR	0.39 ± 0.35	0.98 ± 3.07	0.191	0.044
NWR	0.64 ± 0.10	0.69 ± 0.18	0.211	0.014
PWR	33.88 ± 11.21	37.73 ± 18.21	0.213	0.024
MWR	0.08 ± 0.06	0.23 ± 1.11	0.135	0.160
WHR	0.11 ± 0.21	0.15 ± 0.33	0.160	0.099
SII	914.28 ± 557.78	2494.51 ± 3301.68	0.501	<0.001
ALC	2.22 ± 0.98	1.66 ± 1.35	0.396	<0.001
Hb	120.44 ± 31.35	113.66 ± 32.58	0.277	0.003
LMR	3.78 ± 2.37	2.66 ± 2.14	0.456	<0.001
LWR	0.25 ± 0.09	0.18 ± 0.14	0.375	<0.001
dNLR	0.85 ± 0.06	0.79 ± 0.47	0.125	0.176

**Table 5 medicina-58-01426-t005:** Variable associated with overall survival (OS) when both pre- and post-treatment markers were included in the model.

	Univariate Analysis			Multivariate Analysis *		
Variables	HR	95% CI	*p*-Value	HR	95% CI	*p*-Value
Stage (late vs. early)	2.766	1.509, 5.072	<0.001	2.421	0.999, 5.866	0.0503
Surgery (yes vs. no)	0.364	0.202, 0.657	0.001	0.353	0.138, 0.907	0.0305
LMR pre-tx (high vs. low)	3.426	1.913, 6.134	<0.001	3.057	1.560, 5.990	0.0011
PLC post-tx (high vs. low)	2.905	1.720, 4.906	<0.001	3.346	1.711, 6.544	0.0004
PLR post-tx (high vs. low)	3.384	1.604, 7.120	0.001	5.261	2.615, 10.583	<0.0001

* Cox regression (stepwise model).

**Table 6 medicina-58-01426-t006:** Variables associated with disease free survival (DFS) when both pre- and post-treatment markers were included in the model.

	Univariate Analysis			Multivariate Analysis *		
Variables	HR	95% CI	*p*-Value	HR	95% CI	*p*-Value
Surgery (yes vs. no)	0.388	0.198, 0.762	0.006	0.422	0.183, 0.975	0.0434
SII pre-tx (high vs. low)	2.991	1.840, 4.863	<0.001	2.593	1.500, 4.482	0.0006
PLC post-tx (high vs. low)	2.358	1.404, 3.958	0.001	1.919	1.110, 3.317	0.0196
PLR post-tx (high vs. low)	3.387	1.604, 7.152	0.001	3.441	1.983, 5.969	<0.0001

* Cox regression (stepwise model).

## Data Availability

Not applicable.

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
