# Peer review of "Prognostic Abilities of Pre- and Post-Treatment Inflammatory Markers in Oral Squamous Cell Carcinoma: Stepwise Modelling"

_medicina, 2022, doi:10.3390/medicina58101426_

Round 1

Reviewer 1 Report

This is an interesting paper regarding the prognosis of patients with OSCC.

However, the study needs some validation of the methodology.

Please review the following and add them to the discussion page.

This is an interesting paper regarding the prognosis of patients with OSCC.

However, the study needs some validation of the methodology.

Please review the following and add them to the discussion page.This is an interesting paper regarding the prognosis of patients with OSCC.

However, the study needs some validation of the methodology.

Please review the following and add them to the discussion page.

1. Results of blood sampling in the first postoperative week

 Although the inflammatory response is generally considered to have recovered, there may be some patients who have not fully recovered.

2. Are the surgical procedures consistent?

 Surgeries that involve extensive reconstruction are likely to be lengthy and invasive. This would naturally lead to a longer period of high inflammatory marker values.

3. If you want to verify the OS and DFS, you need to use the Kaplan-Meier method.

It is necessary to verify the above.

Author Response

Dear Reviewer

Thank you for your comments. We have carried out the changes as suggested to the best of our abilities. Kindly see the attached documents for the details. 

Thank you

AR

Reviewer 2 Report

The study is interesting and genuine, however the authors should address the following points to improve the quality of the manuscript:

- The abstract should be non-structured with specific word limits (please refer to the authors' guidelines).

- It is advisable to use passive voice for scientific writing.

- The introduction section needs significant changes to address the outstanding research gap, the current research question, objectives and null hypotheses.

- The methodology section is comprehensive, however the figures should be impeded within the text for clarity and makes it easy to follow.

- The authors should re-organize the whole manuscript (by inserting the figures and tables where they belong) for clarity.

- The discussion part should be formatted as paragraphs only. Comparison with the existing literature, limitations and future directions for research should be included.

- The conclusion section may be summarized in bullets.

Author Response

(The authors gave the same response as above.)

Round 2

Reviewer 1 Report

General discussion of the study

It is inappropriate to argue that these inflammatory markers alone are prognostic factors, even if the limitation states that these factors are not considered, unless the study takes into account the influence of factors related to inflammatory markers, such as the degree of invasiveness of surgery and the progression of the tumor.

Author Response

Dear Reviewer

Thank you for your comments and we agree that the degree of invasiveness of the surgery may be a confounding factor with the inflammatory markers which we have given as one of the limitations of this study (as below). However, as for the progression of the tumour we have taken it into account by analyzing the disease-free survival (DSF) for the included cases.

Furthermore, we have added the following:

Therefore, it is recommended to carry out a multi-centre prospective study with a larger sample size be conducted, taking into account the type of surgical procedures and treatment being administered, the degree of invasiveness and progression of tumour which would be reflected in the stage of the disease and to increase generalizability of the findings and eliminate any risk of bias.

Thank you

AR

Reviewer 2 Report

All the necessary changes have been done.

the manuscript can be accepted in the present form.

Author Response

Dear Reviewer

Thank you for accepting our manuscript for publication.

Thank you

AR